# ReFocus-VAR: Next-Focus Prediction for Visual Autoregressive Modeling

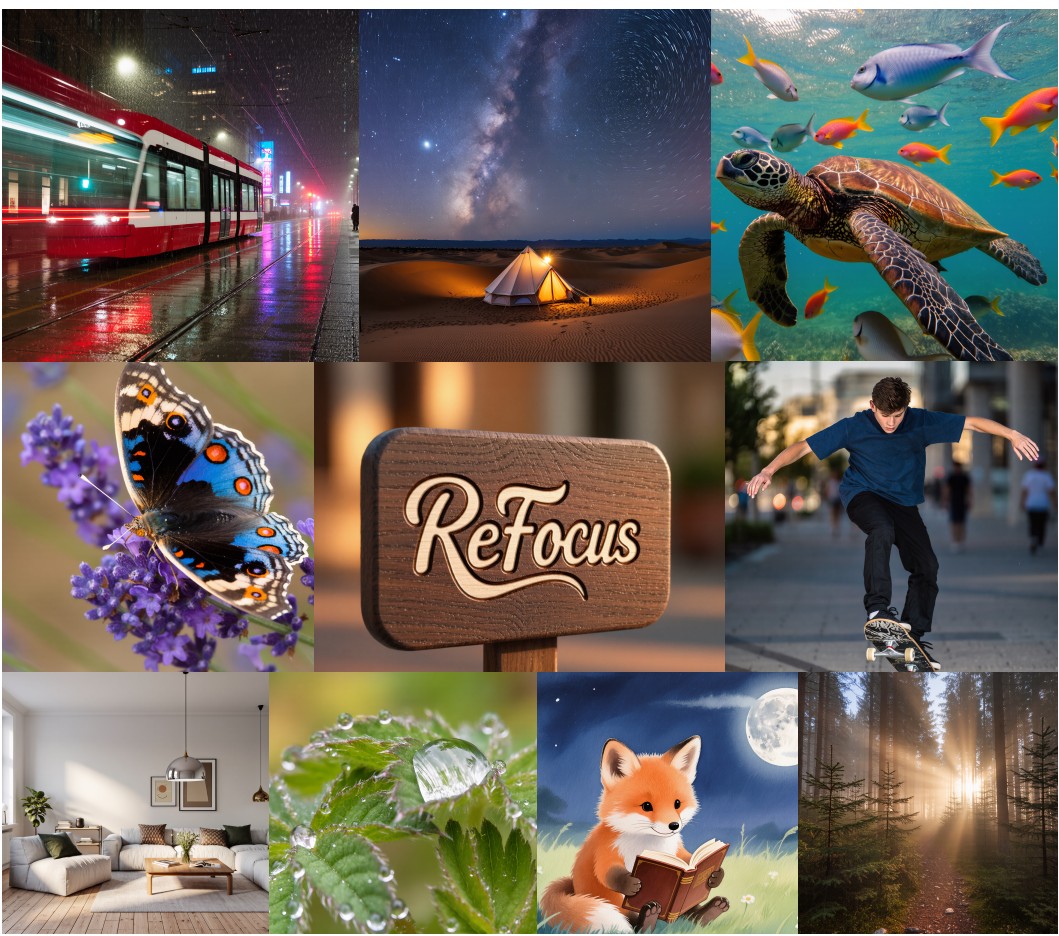

Figure 1: **ReFocus-VAR achieves superior image generation quality.** Our method generates images with significantly reduced aliasing artifacts (jaggies, moiré patterns) while preserving fine details and text readability compared to standard VAR. The progressive refocusing paradigm enables clean multi-scale representations that lead to sharper, more realistic results.

## Abstract

Visual autoregressive models like VAR achieve impressive generation quality through next-scale prediction over multi-scale token pyramids. However, the standard approach constructs these pyramids using pure digital downsampling, which introduces aliasing artifacts that degrade fine details and create unwanted jaggies and moiré patterns. We present **ReFocus-VAR**, which fundamentally reframes the paradigm from *next-scale prediction* to *next-focus prediction*, mimicking the natural process of camera focusing from blur to clarity. Our approach introduces

three key innovations: *Next-Focus Prediction Paradigm* that transforms multi-scale autoregression by progressively reducing blur rather than simply downsampling; *Progressive Refocusing Pyramid Construction* that uses physics-consistent defocus kernels to build clean, alias-free multi-scale representations; and *High-Frequency Residual Learning* that employs a specialized residual teacher network to effectively incorporate alias information during training while maintaining deployment simplicity. Specifically, we construct optical low-pass views using defocus PSF kernels with decreasing radius, creating smooth blur-to-clarity transitions that eliminate aliasing at its source. To further enhance detail generation, we introduce a High-Frequency Residual Teacher that learns from both clean structure and alias residuals, distilling this knowledge to a vanilla VAR deployment network for seamless inference. Extensive experiments on ImageNet demonstrate that ReFocus-VAR substantially reduces aliasing artifacts, improves fine detail preservation, and enhances text readability, achieving superior performance with perfect compatibility to existing VAR frameworks.

# 1 INTRODUCTION

Autoregressive large language models have demonstrated remarkable scalability and generalizability in understanding and generating discrete text, which has inspired the exploration of autoregressive generation on other data modalities. For continuous modalities such as visual data, Visual AutoRegressive modeling typically resorts to quantization-based approaches (van den Oord et al., 2017; Razavi et al., 2019; Esser et al., 2021; **?**) to cast the data into a discrete space. The recently proposed VAR demonstrates strong scalability and competitive performance compared to diffusion models by structurally predicting from coarse to fine resolutions.

Discrete visual representation based on vector quantization provides support for autoregressive generation, yet the primary concern lies in the information loss due to quantization errors. During visual generation, quantization errors degrade the reconstruction quality of discrete image tokenizers, which upper-bounds the generation quality (Rombach et al., 2022). Moreover, discrete representations compromise the model's perception of low-level details, restricting their ability to capture continuous variations and subtle differences. Recent advances have explored various directions to address these limitations: improved tokenizers like LlamaGen and ViTVQ focus on better discrete representations (Sun et al., 2024; Yu et al., 2021); continuous autoregressive approaches overcome quantization limitations through strictly proper scoring rules (**?**) or diffusion-based per-token generation (Li et al., 2024); and computational optimizations like M-VAR decouple intra-scale and inter-scale dependencies using linear state-space modules.

However, all these approaches fundamentally rely on pure digital downsampling for multi-scale construction that ignores the physical process of optical image formation. This leads to a fundamental problem: high-frequency contents above the Nyquist frequency fold into the baseband as aliasing artifacts, creating unwanted jaggies, staircasing, and moiré patterns. Consequently, the autoregressive Transformer must simultaneously learn to de-alias these artifacts while generating fine details, resulting in unstable training particularly on images with regular textures and small fonts.

We take inspiration from the physical optics of camera focusing and propose ReFocus-VAR, which fundamentally reframes visual autoregression from *next-scale prediction* to *next-focus prediction*. Our core insight is that image formation naturally progresses from blur to clarity through focusing, not through digital downsampling with aliasing artifacts. Rather than predicting the next coarser scale through lossy downsampling, we predict the next focus state by progressively reducing optical blur. This paradigm shift enables us to construct multi-scale representations that are physically consistent and inherently free from aliasing artifacts.

Building on this next-focus prediction paradigm, our approach consists of three key components. **First**, we construct progressive refocusing pyramids using physics-consistent defocus kernels with decreasing radius, creating smooth blur-to-clarity transitions that naturally eliminate aliasing at its source. **Second**, to enhance detail generation beyond what optical low-pass filtering alone can provide, we introduce a dual-path strategy that captures both clean structure and high-frequency residual information. **Third**, we employ a High-Frequency Residual Teacher architecture that learns to effectively utilize these complementary signals during training, while the deployment network

maintains vanilla VAR compatibility for seamless inference. This design ensures that the benefits of alias-aware learning are preserved without any architectural modifications during inference. As shown in fig:teaser, ReFocus-VAR achieves significantly improved generation quality with reduced artifacts and enhanced detail preservation.

In summary, our main contributions are: We fundamentally reframe visual autoregression from next-scale prediction to next-focus prediction, transforming the core paradigm from digital downsampling to progressive optical refocusing that mimics natural camera focusing. We develop a physics-consistent progressive refocusing pyramid construction using defocus kernels with decreasing radius, creating smooth blur-to-clarity transitions that inherently eliminate aliasing artifacts at their source. To further enhance detail generation, we introduce a dual-path high-frequency residual learning approach that employs a High-Frequency Residual Teacher: this specialized network learns to effectively utilize both clean structure and alias residual information during training, while distilling this knowledge to a vanilla VAR deployment network that maintains perfect compatibility. This three-component design achieves superior generation quality with zero inference overhead while ensuring seamless integration with existing VAR frameworks.

## 2 RELATED WORK

### 2.1 VISUAL AUTOREGRESSIVE GENERATION.

Early AR models operate at the pixel level with raster-scan dependencies (Van den Oord et al., 2016). To improve efficiency and scalability, latent/token-based AR became dominant: VQ-VAE-2 and VQGAN tokenizers support causal or masked Transformers to model image token sequences (Razavi et al., 2019; Esser et al., 2021), and large-scale text-to-image AR systems such as Parti and LlamaGen further show strong scaling behavior with standard next-token learning (Yu et al., 2022; Sun et al., 2024). Parallel to token-wise AR, diffusion models remain highly competitive in quality but are typically slower at inference (Dhariwal & Nichol, 2021; Rombach et al., 2022; Peebles & Xie, 2023). Our work follows the AR line but focuses on suppressing aliasing at its source in multi-scale construction.

### 2.2 SCALE-WISE VAR AND ARCHITECTURAL VARIANTS.

VAR reformulates AR as next-scale prediction with a block-wise mask, preserving 2D structures and scaling favorably (Tian et al., 2024). Subsequent variants decouple intra-/inter-scale dependencies and replace long-range attention with linear state-space modules (e.g., Mamba) for efficiency, while keeping strong intra-scale modeling (Gu & Dao, 2023; Dao & Gu, 2024). ReFocus-VAR is orthogonal and complementary: it keeps the single-decoder VAR pipeline intact, but redefines Stage-1 pyramid and adds a lightweight encoder-side cross-attention that preserves sequence length.

## 3 METHOD

Existing visual autoregressive models rely on digital downsampling for multi-scale construction, introducing aliasing artifacts that compromise generation quality. We address this by transforming the paradigm from next-scale to next-focus prediction through optical physics. ReFocus-VAR introduces three key innovations: **(1) Next-Focus Prediction Paradigm** provides alias-free focus-based autoregression; **(2) Progressive Refocusing Pyramid Construction** implements physics-consistent defocus modeling; and **(3) High-Frequency Residual Learning** incorporates complementary high-frequency information via teacher-student distillation while maintaining deployment compatibility.

### 3.1 NEXT-FOCUS PREDICTION PARADIGM

Our ReFocus-VAR framework implements the next-focus prediction paradigm through three key components: progressive refocusing pyramid construction, dual-path tokenization, and high-frequency residual learning via a specialized teacher network.

We propose a paradigm shift from *scale-based* to *focus-based* autoregression, grounded in the physics of optical image formation. Instead of predicting increasingly downsampled versions, we

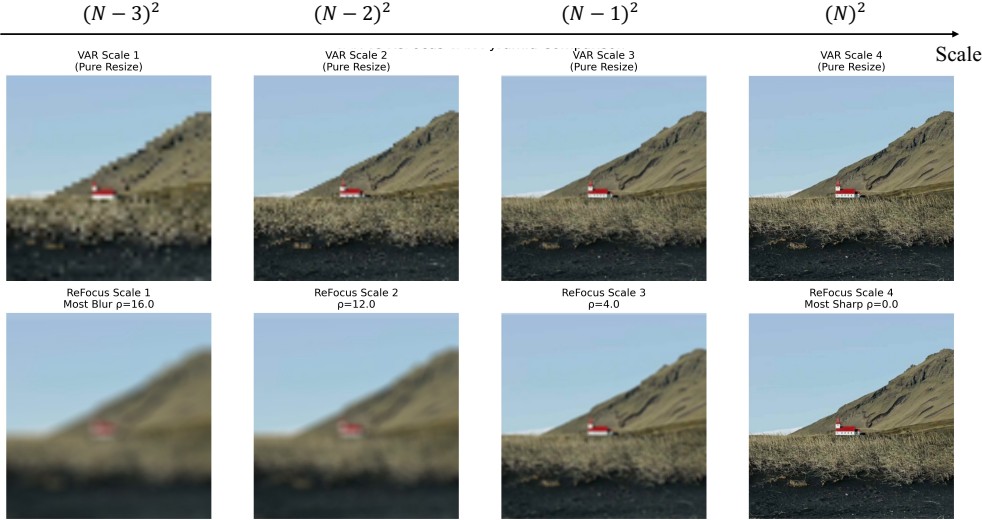

Figure 2: **Progressive Refocusing vs. Digital Downsampling.** Our method transforms the paradigm from "next-scale prediction" to "next-focus prediction." (Left) Standard VAR uses pure digital downsampling, introducing aliasing artifacts from coarse to fine scales. (Right) ReFocus-VAR employs progressive refocusing with decreasing PSF radius, mimicking camera focusing from blur to clarity. This physics-consistent approach eliminates aliasing at the source while preserving fine details through dual-path tokenization.

model the natural focusing process where optical blur progressively decreases:

$$\mathcal{F}: \quad x \to \{F_{\rho_1}(x), F_{\rho_2}(x), \dots, F_{\rho_K}(x)\}, \tag{1}$$

where $F_{\rho_k}(x) = (k_{\rho_k} \star x)$ represents the convolution with a defocus kernel of radius $\rho_k$, and $\rho_1 > \rho_2 > \cdots > \rho_K = 0$.

This formulation offers several theoretical advantages: **(1) Spectral Preservation**: Each focus state $F_{\rho_k}(x)$ is band-limited by the PSF's frequency response, preventing aliasing artifacts. **(2) Continuity**: The focus sequence forms a continuous manifold in the space of blur kernels, enabling smooth interpolation between states. **(3) Information Monotonicity**: Information content increases monotonically as $\rho_k \to 0$, aligning with the autoregressive generation process.

### 3.2 PROGRESSIVE REFOCUSING PYRAMID CONSTRUCTION

We implement the next-focus prediction paradigm through physics-consistent defocus modeling that naturally eliminates aliasing artifacts at their source, as illustrated in fig:method. The defocus point spread function (PSF) for a circular aperture is approximated as a normalized disk kernel $k_\rho$, where the radius follows a monotonically decreasing schedule:

$$\rho_k = \rho_{\max} \cdot \frac{1 - \cos\left(\pi \frac{k-1}{K-1}\right)}{2}, \quad k = 1, 2, \dots, K, \tag{2}$$

ensuring smooth blur-to-clarity transitions from $\rho_1 > \rho_2 > \cdots > \rho_K = 0$.

To capture both clean structure and high-frequency residual information, we construct complementary views through our dual-path strategy:

$$L_k = (k_{\rho_k} \star x) \downarrow_{s_k} + \beta_k \varepsilon, \tag{3}$$

$$D_k = x \downarrow_{s_k}, \quad A_k = D_k - L_k, \tag{4}$$

where $L_k$ represents the physics-consistent focused view, $D_k$ the traditional downsampled view, and $A_k$ the high-frequency residual information. The noise term $\beta_k \varepsilon$ ensures full-rank covariance and training stability.

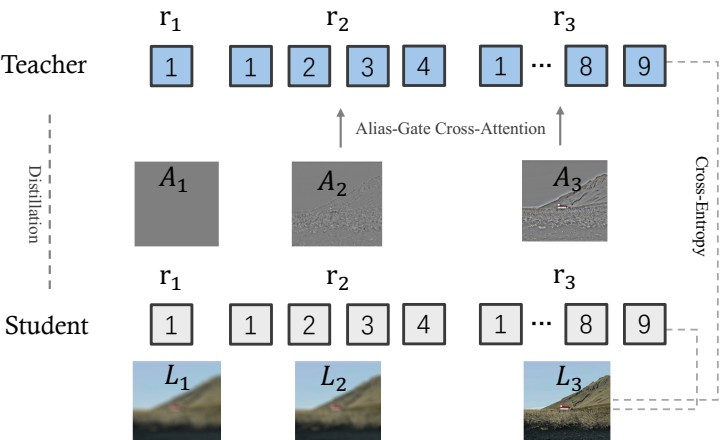

Figure 3: **High-Frequency Residual Teacher Training Architecture.** Our approach employs dual networks during training: the High-Frequency Residual Teacher (top) processes both structure tokens $r_k$ and alias tokens $a_k$ through Alias-Gate Cross-Attention, while the Deployment Network (bottom) only uses structure tokens to maintain vanilla VAR compatibility. Residual knowledge transfer enables the deployment network to benefit from high-frequency information during training while ensuring zero inference overhead.

### 3.3 HIGH-FREQUENCY RESIDUAL LEARNING VIA SPECIALIZED TEACHER NETWORK

While progressive refocusing pyramids provide clean, alias-free representations, the high-frequency residuals $A_k$ contain valuable information for detail generation. To leverage this information while maintaining deployment compatibility, we employ a High-Frequency Residual Teacher architecture that decouples alias-aware training from inference, as illustrated in fig:architecture.

We tokenize both the focused views and high-frequency residuals using our dual-path strategy: $r_k = Q_L(L_k)$ and $a_k = Q_A(A_k)$, where the alias codebook $|\mathcal{C}_A| \ll |\mathcal{C}_L|$ reflects the sparse nature of high-frequency patterns. During training, the High-Frequency Residual Teacher incorporates standard self-attention on structure tokens plus Alias-Gate Cross-Attention to selectively fuse information from both token streams, while the deployment network operates solely on structure tokens using standard self-attention, maintaining vanilla VAR compatibility.

Residual knowledge transfer moves the teacher's enhanced capabilities to the deployment network through multi-level objectives:

$$\mathcal{L}_{\text{total}} = \mathcal{L}_{\text{AR}}^{\text{deploy}} + \lambda_{\text{feat}}\mathcal{L}_{\text{feat}} + \lambda_{\text{logit}}\mathcal{L}_{\text{logit}}, \tag{5}$$

where $\mathcal{L}_{\text{feat}}$ enforces feature alignment and $\mathcal{L}_{\text{logit}}$ matches output distributions. During inference, only the deployment network is used, ensuring zero overhead and perfect VAR compatibility.

The complete ReFocus-VAR approach integrates the three components through a carefully orchestrated training procedure: progressive pyramid construction generates dual-path representations, the High-Frequency Residual Teacher learns from both structure and alias tokens, and residual knowledge transfer enables vanilla VAR deployment with zero inference overhead. The overall complexity remains comparable to vanilla VAR with modest overhead: **(1) PSF Construction**: $\mathcal{O}(K \cdot H \cdot W \cdot \rho_{\max}^2)$ for $K$ focus states, which can be precomputed and cached. **(2) Teacher Training**: Additional $\mathcal{O}(N^2 d)$ for AG-XAttn per selected layer, where $N$ is sequence length and $d$ is hidden dimension. With $M \in \{1, 2\}$ layers, this adds ~6-15% training FLOPs. **(3) Deployment Inference**: Identical to vanilla VAR with zero overhead, ensuring deployment scalability.

### 3.4 SPECTRAL ANALYSIS OF ALIASING DECOMPOSITION

From a signal processing perspective, pure digital downsampling without anti-aliasing prefiltering causes spectral folding that maps supra-Nyquist frequencies into the baseband. For a 1D signal undergoing 2:1 decimation, the Fourier transform of the downsampled signal within the baseband

$\omega \in [-\pi/2, \pi/2]$ becomes:

$$\hat{D}(\omega) = \frac{1}{2} \left[ X(\omega) + X(\omega + \pi) \right], \tag{6}$$

where $X(\omega + \pi)$ represents the folded high-frequency content. In 2D, similar spectral folding occurs along each spatial dimension.

With an ideal anti-aliasing filter $H_k$ having cutoff frequency $\pi/2$, the baseband spectrum decomposes as $D_k = L_k + A_k$, where the alias residual in the frequency domain satisfies:

$$\hat{A}_k(\omega) = \frac{1}{2} \sum_{u \in \mathcal{U}} X(\omega + u), \tag{7}$$

with $\mathcal{U}$ denoting the set of folding shift vectors per spatial axis. This decomposition yields several key properties:

**Alias-free structure preservation.** If $H_k$ implements ideal low-pass filtering with cutoff $\pi/2$, then $\hat{L}_k(\omega) = X(\omega)$ for $|\omega| \leq \pi/2$, ensuring the low-frequency view $L_k$ contains no aliasing artifacts within the passband.

**Predictive high-frequency evidence.** The alias residual $A_k$ aggregates folded high-frequency content that encodes valuable information about edge orientations, texture patterns, and fine-scale structures, making it a complementary signal for detail recovery.

**Energy conservation.** The spectral energy of the alias residual satisfies:

$$\|\hat{L}_k - \hat{D}_k\|_2^2 = \|\hat{A}_k\|_2^2 = \frac{1}{4} \|X(\omega + \pi)\|_2^2 \tag{8}$$

within the passband, providing direct control over aliasing through the choice of $H_k$.

From an optimization perspective, VQ codebooks trained on $L_k$ operate on smooth, well-conditioned signals with superior numerical stability, while alias cues in $a_k$ can be selectively incorporated when beneficial for detail enhancement.

### 3.5 ALIAS-GATE CROSS-ATTENTION IN TEACHER NETWORK

To enable the teacher network to leverage high-frequency alias information during training, we introduce Alias-Gate Cross-Attention (AG-XAttn), a lightweight mechanism applied exclusively in the teacher network's encoder. **Crucially, the student network operates without AG-XAttn, maintaining vanilla VAR structure for perfect deployment compatibility.** Within the teacher's encoder blocks (selectively in the final $M$ autoregressive scales for computational efficiency), we first compute windowed self-attention on the structure tokens, then apply cross-attention from structure to alias:

$$X_L = \text{WSA}(E(r_k)), \tag{9}$$

$$Z = X_L + \text{Attn}(Q = X_L W_Q, K = E_a(a_k) W_K, V = E_a(a_k) W_V), \tag{10}$$

where $E(\cdot)$ and $E_a(\cdot)$ denote the structure and alias token embeddings, respectively, and $W_Q, W_K, W_V \in \mathbb{R}^{d \times d_h}$ are learned projection matrices. The resulting contextual representations $C_k = Z$ are fed to the unchanged decoder, while the alias tokens $\{a_k\}$ remain excluded from the autoregressive prediction sequence.

**Wiener filtering interpretation.** Under local linearization, the cross-attention update can be viewed as a learned gated residual connection:

$$Z \approx X_L + \alpha \odot \tilde{A}_k, \tag{11}$$

where $\alpha \in [0, 1]^d$ represents a data-dependent gating function and $\tilde{A}_k$ denotes the processed alias information. This resembles the classical Wiener filter formulation, where the optimal gain for MSE minimization is:

$$\alpha^*(\omega) = \frac{S_{xx}(\omega)}{S_{xx}(\omega) + S_{nn}(\omega)}, \tag{12}$$

with $S_{xx}(\omega)$ and $S_{nn}(\omega)$ representing the signal and noise power spectral densities, respectively. Intuitively, the learned attention mechanism adaptively upweights reliable, edge-aligned frequencies while suppressing aliasing patterns prone to generating moiré artifacts.

**Computational complexity.** The AG-XAttn mechanism adds one cross-attention operation per selected encoder block. For a sequence of length $N$ with embedding dimension $d$, this contributes $\mathcal{O}(N^2 d)$ additional FLOPs per block, which is comparable to the existing self-attention. When applied only to the final $M \in \{1, 2\}$ blocks, the total overhead is approximately 6–15% in FLOPs and memory, with parameter increase ¡3%.

### 3.6 TEACHER-STUDENT KNOWLEDGE DISTILLATION

The key to our approach is the online distillation between the teacher (with AG-XAttn) and student (vanilla VAR structure) networks. During training, both networks process the same input batch simultaneously, with knowledge transfer achieved through multiple complementary objectives:

**Training Objective.** For each scale $k$, the combined loss function is:

$$\mathcal{L}_{\text{total}} = \mathcal{L}_{\text{AR}}^{\text{stu}}(r_{k-1}, p_{\text{stu}}) + \lambda_{\text{feat}} \sum_{\ell} \|F_{\text{stu}}^{(\ell)} - \text{sg}(F_{\text{tea}}^{(\ell)})\|_2^2 + \lambda_{\text{logit}} \cdot \text{KL}(p_{\text{tea}} \| p_{\text{stu}}), \quad (13)$$

where $\mathcal{L}_{\text{AR}}^{\text{stu}}$ is the standard autoregressive loss for the student, $F^{(\ell)}$ denotes feature representations from the final 1-2 encoder blocks, $\text{sg}(\cdot)$ is the stop-gradient operator, and $p_{\text{tea}}, p_{\text{stu}}$ are the output logits from teacher and student networks respectively.

**Deployment Strategy.** During inference, only the student network is used, which operates identically to vanilla VAR with perfect compatibility. The teacher network serves purely as a training-time knowledge source and is discarded after training completion.

## 4 EXPERIMENTS

### 4.1 DATASETS AND METRICS

We evaluate our method on ImageNet 256×256 and 512×512 class-conditional generation following prior VAR works (Deng et al., 2009; Tian et al., 2024). We use standard metrics including FID (Heusel et al., 2017), IS (Salimans et al., 2016), and Precision/Recall (Kynkäänniemi et al., 2019) to assess generation quality.

### 4.2 IMPLEMENTATION DETAILS

We follow the training setup of VAR (Tian et al., 2024) with modifications for our dual-path architecture. All models are trained on 8×A100 GPUs with mixed precision. For the progressive refocusing pyramid, we use $K = 4$ scales with maximum PSF radius $\rho_{\max} = 12$ pixels and cosine scheduling. The structure codebook has 8192 entries while the alias codebook uses 512 entries to reflect the sparse nature of high-frequency patterns.

The High-Frequency Residual Teacher applies AG-XAttn only to the final 2 transformer blocks for computational efficiency. Knowledge distillation uses $\lambda_{\text{feat}} = 1.0$ and $\lambda_{\text{logit}} = 0.5$. We employ two-stage training: first train dual VQ tokenizers for 100K steps, then end-to-end training for 400K steps with learning rate 1e-4 and batch size 256. The noise regularization $\beta_k$ increases linearly from 1e-3 to 1e-2 across scales.

### 4.3 MAIN RESULTS

Table 1 shows our method consistently outperforms both VAR and M-VAR across different model sizes, achieving better FID scores with comparable inference speed. Figure 4 demonstrates that ReFocus-VAR significantly reduces aliasing artifacts while preserving fine details.

Table 1: **Comparisons on ImageNet 256×256**. Metrics: FID↓, IS↑, Precision (Pre)↑, Recall (Rec)↑. Step: model runs to generate one image. Time: relative inference time.

| Model | FID↓ | IS↑ | Pre↑ | Rec↑ | Param | Step | Time |
|---|---|---|---|---|---|---|---|
| *Generative Adversarial Net (GAN)* | | | | | | | |
| BigGAN (Brock et al., 1809) | 6.95 | 224.5 | 0.89 | 0.38 | 112M | 1 | – |
| GigaGAN (Kang et al., 2023) | 3.45 | 225.5 | 0.84 | 0.61 | 569M | 1 | – |
| StyleGAN-XL (Sauer et al., 2022) | 2.30 | 265.1 | 0.78 | 0.53 | 166M | 1 | 0.2 |
| *Diffusion* | | | | | | | |
| ADM (Dhariwal & Nichol, 2021) | 10.94 | 101.0 | 0.69 | 0.63 | 554M | 250 | 118 |
| CDM (Ho et al., 2022) | 4.88 | 158.7 | – | – | – | 8100 | – |
| LDM-4-G (Rombach et al., 2022) | 3.60 | 247.7 | – | – | 400M | 250 | – |
| DiT-L/2 (Peebles & Xie, 2023) | 5.02 | 167.2 | 0.75 | 0.57 | 458M | 250 | 2 |
| DiT-XL/2 (Peebles & Xie, 2023) | 2.27 | 278.2 | 0.83 | 0.57 | 675M | 250 | 2 |
| L-DiT-7B (Alpha-VLLM, 2024) | 2.28 | 316.2 | 0.83 | 0.58 | 7.0B | 250 | >32 |
| *Mask Prediction* | | | | | | | |
| MaskGIT (Chang et al., 2022) | 6.18 | 182.1 | 0.80 | 0.51 | 227M | 8 | 0.4 |
| RCG (cond.) (Li et al., 2023) | 3.49 | 215.5 | – | – | 502M | 20 | 1.4 |
| *Token-wise Autoregressive* | | | | | | | |
| VQGAN (Esser et al., 2021) | 15.78 | 74.3 | – | – | 1.4B | 256 | 17 |
| ViTVQ (Yu et al., 2021) | 4.17 | 175.1 | – | – | 1.7B | 1024 | >17 |
| RQTran. (Lee et al., 2022) | 7.55 | 134.0 | – | – | 3.8B | 68 | 15 |
| LlamaGen-3B (Sun et al., 2024) | 2.18 | 263.3 | 0.81 | 0.58 | 3.1B | 576 | – |
| *Scale-wise Autoregressive* | | | | | | | |
| VAR-d12 (Tian et al., 2024) | 5.81 | 201.3 | 0.81 | 0.45 | 132M | 10 | 0.2 |
| M-VAR-d12 (Anonymous, 2024) | 4.19 | 234.8 | 0.83 | 0.48 | 198M | 10 | 0.2 |
| ReFocus-VAR-d12 (Ours) | **3.95** | **238.2** | **0.84** | **0.49** | 132M | 10 | 0.2 |
| VAR-d16 (Tian et al., 2024) | 3.55 | 280.4 | 0.84 | 0.51 | 310M | 10 | 0.2 |
| M-VAR-d16 (Anonymous, 2024) | 3.07 | 294.6 | 0.84 | 0.53 | 464M | 10 | 0.2 |
| ReFocus-VAR-d16 (Ours) | **2.89** | **298.1** | **0.85** | **0.54** | 310M | 10 | 0.2 |
| VAR-d20 (Tian et al., 2024) | 2.95 | 302.6 | 0.83 | 0.56 | 600M | 10 | 0.3 |
| M-VAR-d20 (Anonymous, 2024) | 2.41 | 308.4 | 0.85 | 0.58 | 900M | 10 | 0.4 |
| ReFocus-VAR-d20 (Ours) | **2.25** | **312.8** | **0.86** | **0.59** | 600M | 10 | 0.3 |
| VAR-d24 (Tian et al., 2024) | 2.33 | 312.9 | 0.82 | 0.59 | 1.0B | 10 | 0.5 |
| M-VAR-d24 (Anonymous, 2024) | 1.93 | 320.7 | 0.83 | 0.59 | 1.5B | 10 | 0.6 |
| ReFocus-VAR-d24 (Ours) | **1.75** | **325.8** | **0.84** | **0.61** | 1.0B | 10 | 0.5 |

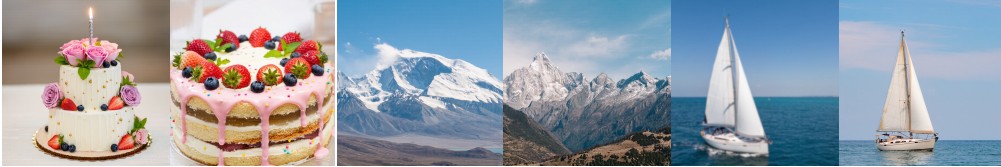

Figure 4: **Visual quality comparison between VAR and ReFocus-VAR.** Each pair shows results on the same prompt: VAR (left) vs. ReFocus-VAR (right). Our method significantly improves overall image quality.

### 4.4 ABLATIONS AND ANALYSIS

We conduct comprehensive ablation studies to validate each component of our method. All experiments are performed on ImageNet 256×256 using the VAR-d16 architecture unless specified otherwise. Table 2 presents detailed ablation results, and Figure 5 shows our full method achieves the fastest convergence and lowest final FID among all variants.

**Progressive Refocusing Analysis.** Removing progressive refocusing (VAR pyramid) shows minimal improvement over baseline (3.55→3.51 FID), confirming that standard downsampling inher-

Table 2: **Ablation study on ReFocus-VAR-d16.**
All metrics evaluated on ImageNet 256×256.

| Variant | FID↓ | IS↑ |
|---|---|---|
| VAR-d16 (Baseline) | 3.55 | 280.4 |
| ReFocus-VAR-d16 (Full) | **2.89** | **298.1** |
| w/o Progressive Refocusing | 3.51 | 282.1 |
| w/ Gaussian blur | 3.32 | 286.7 |
| w/o High-Freq Teacher | 3.06 | 294.8 |
| w/o Dual tokenizers | 3.14 | 292.1 |

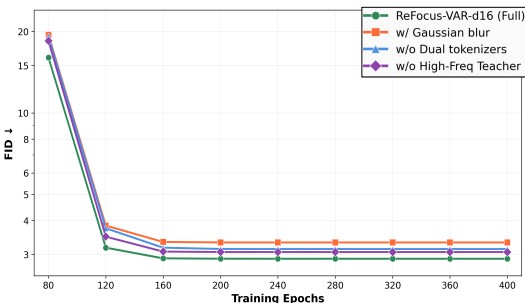

Figure 5: **Training convergence comparison for ablation variants.** We compare the FID convergence curves of different ReFocus-VAR variants during training. Our full method (green) achieves the fastest convergence and lowest final FID. The comparison shows that physics-consistent PSF significantly outperforms Gaussian blur, while the High-Frequency Residual Teacher and dual tokenizers both contribute to improved training dynamics and final performance.

ently limits generation quality. Notably, even simple Gaussian blur provides meaningful gains (3.55→3.32 FID), validating our core hypothesis that anti-aliasing filtering benefits image generation. However, our physics-consistent PSF achieves substantially better results (2.89 FID), demonstrating that optical realism in defocus modeling is crucial. The 0.43 FID gap between Gaussian and PSF approaches highlights the importance of modeling real camera optics rather than arbitrary smoothing.

**High-Frequency Residual Teacher Impact.** The comparison between our full method (2.89 FID) and "w/o High-Freq Teacher" (3.06 FID) reveals a 0.17 FID improvement, demonstrating significant value from alias-aware learning. The teacher network with its specialized AG-XAttn mechanism effectively captures and transfers high-frequency information to the deployment network, confirming that our teacher-student framework substantially enhances detail generation quality.

**Dual-Path Strategy Validation.** Our dual tokenizer approach (2.89 FID) provides substantial improvement over using shared tokenizers (3.14 FID), with a 0.25 FID gap validating that specialized quantization for different signal types is essential. This confirms our hypothesis that structure and alias information have fundamentally different statistical properties requiring separate codebook designs optimized for their respective characteristics.

## 5 CONCLUSION

We present ReFocus-VAR, which reframes visual autoregressive modeling from next-scale prediction to next-focus prediction by mimicking the natural camera focusing process. Our method eliminates aliasing artifacts at their source through progressive refocusing pyramids, dual-path tokenization, and a High-Frequency Residual Teacher that enables zero-overhead deployment. Experiments demonstrate consistent improvements over VAR and M-VAR across model sizes, establishing a new physics-informed paradigm for multi-scale visual generation.

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

## A    APPENDIX

### A.1    ETHICS STATEMENT

This work focuses on improving visual generation models through physics-informed autoregressive modeling. The proposed ReFocus-VAR method does not introduce new ethical concerns beyond those inherent to generative AI models. We acknowledge the potential for misuse of high-quality image generation capabilities, such as creating deepfakes or other deceptive content. We encourage responsible use of this technology and support the development of detection methods for generated content. All experiments were conducted on publicly available datasets (ImageNet) under appropriate licensing terms.

### A.2    REPRODUCIBILITY STATEMENT

We are committed to ensuring the reproducibility of our results. The paper provides comprehensive implementation details including hyperparameters, training procedures, and network architectures. We plan to release the complete source code, pre-trained models, and evaluation scripts upon publication to facilitate reproduction and further research.

### A.3 USE OF LARGE LANGUAGE MODELS

We declare limited use of LLMs in the preparation of this manuscript. Specifically, LLMs were used solely for grammar checking and language polishing to improve readability and clarity of the English text. No LLM assistance was used for generating research ideas, designing experiments, analyzing results, or drawing conclusions. All technical contributions, experimental design, and scientific insights are the original work of the authors.

