# OpenReview forum: "ReFocus-VAR: Next-Focus Prediction for Visual Autoregressive Modeling"
_ICLR.cc/2026/Conference — ICLR 2026 Conference Withdrawn Submission_

### Official Review · Reviewer_3Sd6 · 2025-10-30

**Soundness:** 2
**Presentation:** 3
**Contribution:** 3
**Rating:** 4
**Confidence:** 4

**Summary:**

This paper proposes a novel visual autoregressive generation paradigm called Next-Focus Prediction, aiming at solving the aliasing artifact problem caused by digital downsampling in traditional VAR models. Borrowing from the blur-to-sharp focusing principle of cameras, the authors use optically consistent defocus kernels to progressively reconstruct the image pyramid, achieving a gradual generation from blur to sharp. In ImageNet-like conditional generation experiments, ReFocus-VAR significantly outperforms VAR and M-VAR in metrics such as FID and IS, and effectively removes jagged edges and moiré patterns.

**Strengths:**

1. This work identifies a limitation of traditional VAR models, where digital downsampling can cause aliasing artifacts, and proposes a novel method to effectively address this issue.

2. The proposed Next-Focus Prediction concept is inspired by the camera focusing process in optical photography. This approach is interesting and makes sense.

3. Comprehensive experiments demonstrate the effectiveness of the proposed method. With comparable parameter counts, ReFocus-VAR achieves a lower FID score than the original VAR model.

4. The ablation study confirms the contribution of each proposed component to the overall performance.

**Weaknesses:**

1. I am highly confused by Figure 1 on the first page of the article. It is obvious that this image was not generated by an ImageNet class-conditional model but rather by a text-to-image model. However, the entire paper only presents experimental results for class-conditional generation, with no explanation provided on how Figure 1 was generated. Please clearly clarify the generation process of Figure 1, as this is a critical issue.

2. The paper lacks essential visual comparisons. Since the authors claim that the proposed method can resolve aliasing artifacts, visual comparisons with the original VAR should be included to highlight the differences in this aspect.

3. There are some citation errors in the Introduction section, which require more careful review.

4. The formatting of Table 3 and Figure 4 in the paper needs to be adjusted.

**Questions:**

Please see the weakness. I would change the score according to the response to the problems.

---

### Official Review · Reviewer_gmmx · 2025-10-30

**Soundness:** 3
**Presentation:** 2
**Contribution:** 3
**Rating:** 4
**Confidence:** 5

**Summary:**

* This paper addresses critical limitations of existing visual autoregressive models (e.g., VAR), which rely on pure digital downsampling for multi-scale representation construction—leading to aliasing artifacts (jaggies, moiré patterns), fine-detail loss, and poor text readability. To solve these issues, it proposes ReFocus-VAR, a framework that shifts the paradigm from "next-scale prediction" to "next-focus prediction," mimicking the natural physical process of camera focusing from blur to clarity. ReFocus-VAR integrates three core components to achieve alias-free, high-fidelity image generation while maintaining full compatibility with existing VAR frameworks and zero inference overhead.

* Technically, ReFocus-VAR first constructs a Progressive Refocusing Pyramid using physics-consistent defocus kernels with decreasing radii, creating smooth blur-to-clarity transitions that eliminate aliasing at its source. Second, it adopts a dual-path tokenization strategy to process both physics-aligned focused views and high-frequency residual views. Third, it leverages a High-Frequency Residual Teacher Network to fuse structural and alias information during training, then distills this knowledge into a vanilla VAR deployment network via multi-objective loss.

**Strengths:**

1. The paper is built on a clear and compelling motivation. It addresses a well-known, yet often overlooked, issue in VAR models: the aliasing artifacts caused by naive downsampling. This provides a solid foundation for the work.

2. The proposed method is elegant in its simplicity and can be seamlessly integrated into the standard VAR framework as a drop-in replacement. The empirical results are convincing, showing that this approach leads to significant performance gains over the baseline VAR model while using the same network architecture.

3. I found the analysis in the "SPECTRAL ANALYSIS OF ALIASING DECOMPOSITION" section to be particularly insightful. This theoretical treatment effectively clarifies the nature of aliasing artifacts and provides a solid foundation for the paper's core claims.

**Weaknesses:**

* Incomplete Experimental Analysis: A major methodological flaw is the absence of an evaluation of the teacher model's standalone performance. The paper's narrative relies on a teacher-student distillation process, where the teacher guides the training of the final, efficient student model. However, the performance of this teacher model is never reported. This is a critical baseline, as it represents the performance upper bound for the student. Without it, the reader cannot judge the efficacy of the distillation process itself or quantify the trade-off being made between performance and efficiency. This omission leaves a significant gap in the paper's experimental validation.

* Lack of Clarity and Polish: The paper appears to be in a preliminary state and is not ready for publication. Throughout the manuscript, there are numerous typographical errors, ambiguous phrases, and broken cross-references (e.g., "fig:architecture" on L242). These issues significantly impede readability and suggest a lack of careful preparation. While the core ideas are interesting, the current presentation does not meet the quality standards of a top-tier conference like ICLR. A major revision would be necessary to address these clarity issues.

**Questions:**

* In lines 180-182, corresponding to the caption for Figure 2, it says: "(Left) Standard VAR uses pure digital downsampling, introducing aliasing artifacts from coarse to fine scales. (Right) ReFocusVAR employs progressive refocusing with decreasing PSF radius, mimicking camera focusing from blur to clarity". Is there a typo here? Should "Left" be "Top" and "Right" be "Bottom"?

* Since VAR is a class-conditional image generation model, it cannot generate images based on text. So, how were the images in Figure 1 generated? These visual results do not look like typical ImageNet generation results.

* In line 356 of the paper, it says: "We evaluate our method on ImageNet 256×256 and 512×512 class-conditional generation following prior VAR works." However, I cannot find any experimental results for 512×512 throughout the entire paper.

* In line 242, the text says, "we employ a High-Frequency Residual Teacher architecture that decouples alias-aware training from inference, as illustrated in fig:architecture." Is the figure reference broken here? Is it supposed to point to Figure 3?

* Regarding the statement $L_k$ represents the physics-consistent focused view, theoretically, it is necessary to apply a defocus kernel transform to each scale to form new images. However, the VAR tokenizer is trained on clear images and has not been exposed to this distribution. Does this lead to a significant domain drift problem?

---

### Official Review · Reviewer_Cjtq · 2025-10-31

**Soundness:** 3
**Presentation:** 3
**Contribution:** 3
**Rating:** 4
**Confidence:** 4

**Summary:**

this paper reframes scale-wise visual autoregressive (VAR) generation as “next-focus” prediction. Instead of building multi-scale pyramids by digital downsampling (which can introduce aliasing), the method constructs a progressive refocusing pyramid using physics-motivated defocus PSFs with decreasing radii, then trains with a dual-path tokenizer (structure vs. alias residuals) and a “High-Frequency Residual Teacher” that uses an Alias-Gate Cross-Attention module to fuse residual cues during training while keeping deployment identical to vanilla VAR. On ImageNet, the authors report consistent gains (e.g., VAR-d16 FID 3.55 → 2.89; IS 280.4 → 298.1) with “zero” inference overhead and modest extra FLOPs during training (≈6–15%), along with ablations suggesting PSF-based refocusing outperforms Gaussian blur and that the teacher and dual tokenizers each contribute to the improvement.

while the framing is appealing, much of the benefit seems attributable to applying an anti-aliasing prefilter—a well-known fix in multirate signal processing and pyramid construction—rather than a fundamentally new learning principle; the paper would be stronger by isolating how much comes from the PSF schedule alone versus architectural/training bells and whistles (teacher, dual tokenizers, AG-XAttn). The reliance on synthetic circular-aperture PSFs raises questions about robustness to real optics (aperture shapes, aberrations, sensor MTF), and experiments are limited to ImageNet class-conditional generation with FID/IS/Prec/Rec; there’s no evaluation on text legibility benchmarks or texture-heavy datasets where aliasing is most problematic. The “zero inference overhead” claim is accurate for the student but the total training cost (extra tokenizers, parallel teacher forward passes) isn’t quantified wall-clock-wise, and comparisons omit recent strongest diffusion/AR baselines at similar compute. Finally, key engineering choices (e.g., why K=4, why ρ_max=12, sensitivity to schedules/codebook sizes) are only lightly justified, leaving open whether the gains are broadly reproducible or hyperparameter-fragile.

**Strengths:**

Recasting scale-wise VAR as next-focus prediction is a crisp idea; the progressive refocusing pyramid is motivated by optical image formation rather than ad-hoc downsampling. The paper articulates spectral preservation, continuity of blur states, and monotonic information increase as theoretical advantages.

**Weaknesses:**

Their own ablation shows that simply adding Gaussian blur already yields most of the improvement (FID 3.55→3.32), while removing “progressive refocusing” barely helps over baseline (3.55→3.51). This weakens the novelty/necessity of the physics-based PSF stack relative to a simpler anti-alias fix.

Quantitive experiments are on ImageNet (class-conditional) at 256/512 with standard FID/IS/Precision/Recall; lack results of broader tasks (e.g., text-to-image), limiting claims of generality. Why visualization shows this paper go beyond ImageNet but qualitative results are limited by ImageNet?

The teacher with AG-XAttn introduces ~6–15% extra FLOPs/memory and requires dual tokenizers plus a two-stage training recipe, but the paper provides no wall-clock training time or resource comparison versus baselines

**Questions:**

Why visualization shows this paper go beyond ImageNet but qualitative results are limited by ImageNet?

---

### Official Review · Reviewer_PS3K · 2025-11-01

**Soundness:** 3
**Presentation:** 2
**Contribution:** 2
**Rating:** 2
**Confidence:** 4

**Summary:**

This paper proposes ReFocus-VAR, a physics-inspired improvement to visual autoregressive (VAR) models that replaces the traditional next-scale prediction paradigm with next-focus prediction, mimicking how a camera gradually refocuses from blur to clarity. By constructing progressive refocusing pyramids using defocus PSF kernels instead of digital downsampling, and introducing a High-Frequency Residual Teacher with dual-path tokenization to learn alias-aware details, ReFocus-VAR effectively removes aliasing artifacts (e.g., jaggies, moiré) while preserving fine structures. Experiments on ImageNet show consistent FID and IS improvements over VAR and M-VAR across multiple model sizes, achieving sharper, more realistic generations without adding inference overhead.

**Strengths:**

1. The paper presents a clear paradigm shift from next-scale prediction to next-focus prediction, inspired by physical camera optics, which is interesting and intuitive.

2. Despite introducing new components during training (dual-path encoding, teacher network), the final deployment retains exact VAR architecture and inference cost.

3. Performance on ImageNet-256 is good. It outperforms VAR and M-VAR in terms of FID and IS.

**Weaknesses:**

1. The experiments are only conducted on ImageNet at 256×256 resolution, which raises concerns about the model’s scalability and robustness to higher-resolution inputs. Results on ImageNet-512 or other high-resolution benchmarks are needed to validate the claimed generality.

2. In Table 1, the column “Time: relative inference time” is confusing—it's unclear what baseline the relative speed is measured against, and the table lacks a reference entry with Time = 1 for normalization. This omission makes it difficult to interpret the reported efficiency.

3. The proposed model introduces several new components—such as the High-Frequency Residual Teacher, Alias-Gate Cross-Attention, and dual-path tokenization—yet the paper does not provide quantitative analysis of their additional training cost. Without clear disclosure of computational overhead, the true efficiency and contribution of the method remain questionable.

**Questions:**

The paper has many formatting issues, for example:

1. Wrong citations in line 75 and line 86

2. Typo in line 332

Why is the related work section so short? I think the authors need to do more literature reviews.

---

### Note · Authors · 2025-11-12

**Comment:**

Due to limited time, the paper does not distinguish between large-scale generalization training and experiments on small datasets, and there are also formatting issues.

**Withdrawal Confirmation:**

I have read and agree with the venue's withdrawal policy on behalf of myself and my co-authors.